# Opioid Use Disorder and Alternative mRNA Splicing in Reward Circuitry

**DOI:** 10.3390/genes13061045

**Published:** 2022-06-10

**Authors:** Spencer B. Huggett, Ami S. Ikeda, John E. McGeary, Karla R. Kaun, Rohan H. C. Palmer

**Affiliations:** 1Behavioral Genetics of Addiction Laboratory, Department of Psychology at Emory University, Atlanta, GA 30322, USA; shugget@emory.edu (S.B.H.); ami.ikeda@emory.edu (A.S.I.); 2Providence Veterans Affairs Medical Center, Department of Psychiatry and Human Behavior, Brown University, Providence, RI 02908, USA; john_mcgeary@brown.edu; 3Department of Neuroscience, Brown University, Providence, RI 02912, USA; karla_kaun@brown.edu

**Keywords:** opioid use disorder, spliceosome, BIN1, GWAS/RNA-seq, addiction, psychiatry

## Abstract

Opiate/opioid use disorder (OUD) is a chronic relapsing brain disorder that has increased in prevalence in the last two decades in the United States. Understanding the molecular correlates of OUD may provide key insights into the pathophysiology of this syndrome. Using publicly available RNA-sequencing data, our study investigated the possible role of alternative mRNA splicing in human brain tissue (dorsal–lateral prefrontal cortex (dlPFC), nucleus accumbens (NAc), and midbrain) of 90 individuals with OUD or matched controls. We found a total of 788 differentially spliced genes across brain regions. Alternative mRNA splicing demonstrated mostly tissue-specific effects, but a functionally characterized splicing change in the clathrin and AP-2-binding (CLAP) domain of the Bridging Integrator 1 (*BIN1*) gene was significantly linked to OUD across all brain regions. We investigated two hypotheses that may underlie differential splicing in OUD. First, we tested whether spliceosome genes were disrupted in the brains of individuals with OUD. Pathway enrichment analyses indicated spliceosome perturbations in OUD across brain regions. Second, we tested whether alternative mRNA splicing regions were linked to genetic predisposition. Using a genome-wide association study (GWAS) of OUD, we found no evidence that DNA variants within or surrounding differentially spliced genes were implicated in the heritability of OUD. Altogether, our study contributes to the understanding of OUD pathophysiology by providing evidence of a possible role of alternative mRNA splicing in OUD.

## 1. Introduction

Opiate and opioid use lead to over 70,000 overdose deaths each year in the USA [1]. Opioid use disorder (OUD) is a chronic relapsing syndrome described by 11 clinical symptoms, such as tolerance, withdrawal, and continued use despite negative consequences.

Opioid exposure and OUD are linked to widespread gene expression changes in the brain [2,3]. Changes in gene expression can be more complex than simply increasing or decreasing levels of a particular gene, and instead involve switching in the transcript isoform of that gene being expressed [4].

A single gene has the potential to code for multiple gene isoforms in a process called alternative splicing. Alternative mRNA splicing combines or excludes certain regions of a gene resulting in the possible altered structure and function of a gene’s protein product [5,6,7,8]. Alternative splicing is predominant in brain cells and aberrant alternative splicing is a major contributor to neurological disease [9].

Splicing in the brain shows robust effects with psychiatry [10], but has rarely been explored in addiction. In the few cases where it has been studied, splicing variants have been shown to have a profound impact on addiction-related phenotypes. For instance, alternative splice variants of the dopamine-2-receptor (*DRD2*) are associated with alcohol [11], cocaine [12,13,14], heroin [15,16], and methamphetamine [17] addiction. Similarly, alternative splice variants of the mu-opioid receptor gene (*OPMR1*) have been associated with heroin seeking [18]. Extensive splicing changes also occur across brain circuits associated with addiction in individuals with alcohol use disorder [19].

Various mechanisms may explain transcriptomic changes observed in the brain. One theory suggests that alternative splicing observed in the brain is due to genetic variation [20]. In support of this, DNA variants within or surrounding differentially spliced genes are linked to the heritability of neurodegenerative [21], psychiatric [20], and drug use disorders [22]. Additionally, the aberrant activity of the spliceosome, a group of > 100 genes that underpin alternative mRNA splicing, could explain disrupted splicing in neurological disease. In the brain, spliceosome genes demonstrated enrichment for gene expression associations with cocaine use disorder [23], but not alcohol use disorder [19]. Additional research is needed to better understand the mRNA brain pathology of addiction and OUD.

The current study sought to characterize the mRNA alternative splicing pathology of OUD in mesocorticolimbic brain circuitry. First, we hypothesized that different abundances of gene isoforms would be present between individuals with OUD and the controls across the brain. Further, we hypothesized that spliceosome pathway genes would be enriched in differential expression results of OUD vs. controls and that DNA variants within and around transcriptomic changes in the brain would contribute to the genetic pathology of OUD.

## 2. Methods

### 2.1. Post-Mortem Brain Samples

We used publicly available RNA-sequencing (RNA-seq) data on human OUD from the Sequence Read Archive in the dorsal–lateral prefrontal cortex (dlPFC; *n* = 40), nucleus accumbens (NAc; *n* = 40; SRP319708) [2], and the ventral midbrain (*n* = 50; atlas plates 51–56–dopamine-enriched regions; SRP163130) [3]. All samples were ascertained via routine autopsies in the USA and dlPFC and NAc samples came from the same individuals. Individuals with OUD met DSM-5 diagnostic criteria, died from opioid use overdose intoxication, or abuse (oftentimes combined with other drugs), and had a history of opioid misuse as well as a positive screen of opioids (fentanyl, heroin, oxycodone, etc.) at the time of death. Controls were matched on demographics, technical features, reported no history of opioid exposure or misuse, tested negative for opioids and other drugs of abuse, and mostly died from cardiovascular disease/events (see Table 1). Human brain samples were de-identified and research procedures were approved by the University of Pittsburgh Committee for Oversight of Research and Clinical Training Involving Decedents as well as the Institutional Review Board for Biomedical Research (IRB Project ID no. CR19080015-011).

### 2.2. Sample and Data Processing

We utilized short-read paired-end RNA-seq data that were sequenced via Illumina NextSeq500 (TruSeq Stranded mRNA High Throughput Sample Prep Kits; Illumina, Sand Diego, CA, USA). Paired-end data are critical for alternative splicing analyses as they accurately map reads, detect splice junctions, and can lead to the correct estimation of splice isoforms for over 90% of human genes [24]. Ideally, one would sequence an entire gene from start to finish to assess all its splicing events. However, splicing results that leverage RNA-seq will rely on some modeling assumptions, data processing, and analytical procedures. We performed quality control on the RNA-seq data using Trimmomatic (v 0.39) [25], which removed low quality reads and Illumina adapters (parameters = TruSeq3-PE-2.fa:2:30:10:2 LEADING:3 TRAILING:3 SLIDINGWINDOW:4:20 MINLEN:36). We then aligned RNA-seq reads to the human reference genome (hg19) using STAR (v 2.79; Alexander Dobin, Cold Spring Harbor, NY; parameters = --outSAMstrandField intronMotif,-twopassMode Basic) [26]. Over 91% of reads were aligned to the reference genome across all samples. RNA-seq reads were counted using featureCounts [27]. We found no evidence of poor data quality (see Appendix A). Differential expression analyses filtered out lowly expressed transcripts (>10 read counts per transcript in at least 80% of samples), which resulted in a total of 13,369–21,002 transcripts.

To control for potential confounds of ancestry, our analyses controlled for ancestral principal components. DNA variants from RNA-seq samples were called using *samtools mpileup.* Using bcftools, we filtered low-quality variant calls (Phred score < 20, read depth < 5) as well as filtered variants with low minor allele frequencies (MAF < 10%) that were not in the Hardy–Weinberg Equilibrium (HWE < 1 × 10^−6^; all data). After calling DNA variants, we merged our sample data with the 1000 Genomes reference sample to perform strand alignment, remove palindromic variants, and restrict data to biallelic single nucleotide polymorphisms that were present in the 1000 Genomes (Phase 3; all populations). Ancestral principal components (PCs) were then computed using *flashPCA* [28].

### 2.3. Analyses

#### 2.3.1. Differential Splicing

To investigate differential spliced genes between individuals with a substance use disorder and controls, we used Leafcutter (v 0.2.9; Yang Li, Stanford, CA, USA)–a powerful method that circumvents methodological limitations of previous approaches (e.g., relative isoform or exon usage) [29]. A differentially spliced gene encompasses multiple clusters of individual alternative mRNA splicing events characterized by the Vertebrate Alternative Splicing and Transcription Database (https://vastdb.crg.eu/wiki/Main_Page; accessed on 22 November 2021). Splicing events were categorized into four general categories: (1) exon skipping, (2) intron retention, (3) alternative acceptor donor splice sites, or (4) alternative donor splice sites. Differential splicing was quantified using a Dirichlet-multinomial generalized linear regression and used default Leafcutter filtering parameters (removing clusters: with <5 samples per intron and clusters with <20 reads per group; 9238–11,821 genes post-filtering). The effect size for differential splicing is a percent change spliced in (ΔPSI), which estimates the alternative exon usage between cases and controls. In our analyses, a positive ΔPSI would indicate that an individual with OUD would be more likely to possess a certain protein-coding region in a gene than a control. A differentially spliced gene was required to survive a Benjamini–Hochberg false discovery rate correction (FDR < 0.10) and surpass an effect size-based cut-off (|ΔPSI| > 0.025).

#### 2.3.2. Differential Expression

To examine one potential mechanism underlying differential splicing; we investigated the aberrant expression of spliceosome genes. First, we examined whether spliceosome genes were differentially expressed (FDR < 0.10 and |log_2_ fold change| > 0.5), and then we performed gene set enrichment analyses (GSEAs) [30], which leverage the pre-ranked pathway enrichment test (*fgseaMultilevel*) [31] to determine whether the spliceosome pathway is perturbed in OUD. Enrichment analyses of differentially expressed genes determine whether genes that pass a significance threshold are more likely to be over-represented in a particular pathway. Similarly, the GSEA estimates gene set over-representation in a pathway but instead of using a *p*-value threshold, it walks down a ranked list of genes to determine the point in the data that maximally deviates from zero. Genes surpassing this maximal deviation point are strongly correlated with the trait and dubbed “leading edge”, which drive the GSEA pathway enrichment

All RNA-seq analyses controlled for age, sex, three ancestral PCs, RNA integrity, and brain pH. All covariates were standardized within the sample to aid in model estimation. Results from differential splicing analyses were functionally annotated with EnrichR [32]. We also benchmarked our differential expression findings to previously reported findings [2,3].

#### 2.3.3. Partitioned Heritability

Lastly, we investigated whether genetic factors could explain the differential expression and splicing findings from our analyses. To test this, we used genome-wide association study (GWAS) summary statistics on OUD [33]. This sample included 79,729 individuals of European ancestry (10.70% cases; all controls used opioids). We examined the extent to which DNA variants (single nucleotide polymorphisms) within and around genes and splicing clusters significantly contributed to the known SNP-heritability of OUD. We tested the enrichment of the known OUD SNP-heritability using three gene sets: (1) differentially expressed genes, (2) differentially spliced genes, and (3) spliceosome genes. Roughly 80% of cis-e/sQTLs occur within 100 kilobase (kb) of a gene [34]. Thus, we tested DNA variants within 100 kb of the transcription start and end sites of a gene in our genetic enrichment tests. Our partitioned heritability analyses used the linkage disequilibrium score regression [35], which calculates whether DNA variants from an a priori gene set account for a significant proportion of the known SNP-heritability relative to the other DNA variants outside of the a priori gene set.

## 3. Results

We found 788 differentially spliced genes, including 1788 splicing events associated with OUD across the brain (see Figure 1; Appendix A). The most frequent differential splicing events were exon skipping (51.8%) and intron retention (36.6%). Differentially spliced genes were enriched for a multitude of functions, including: protein phosphorylation, actin cytoskeleton, Schwann cell differentiation, post-synaptic density, and synaptic vesicle endocytosis (see Figure 2). Eight of the differentially spliced genes were present in the KEGG morphine addiction pathway (*ADCY1*, *GABBR1*, *GABRG1*, *GNAI1*, *GNB5*, *GNG10*, *PDE8B,* and *GRK2*). Five differentially spliced genes were present across all tested brain regions (*SNHG14*, *HERC1*, *HILPDA*, *METTL2B,* and *BIN1*), and only *BIN1* corresponded to a functionally characterized splicing event (see Figure 3; see Appendix A).

Next, we performed differential expression analyses. Our results were consistent with previous analyses of these data. Log_2_ fold change estimates were strongly correlated with previously reported differentially expressed genes with these data (r = 0.60–0.98, all *p* < 2 × 10^−16^). In total, we observed 922 differentially expressed transcripts (see Appendix A); of these, only 3.2% were also differentially spliced. No spliceosome gene was differentially expressed between individuals with OUD and the controls (see Figure 4). However, when performing a ranked-based enrichment test, we found that spliceosome genes were significantly upregulated in OUD in the dlPFC, NAc, and the midbrain (all normalized enrichment score = 1.30–2.32, all log_2_error = 0.21–0.78, all *p* = 0.056–2.839 × 10^−9^). The expressions of spliceosome genes were consistent across brain regions and samples (see Figure 5). Eighteen leading-edge genes consistently drove the enrichment of the spliceosome across and demonstrated consistent patterns across individuals (see Appendix A).

Lastly, we investigated the potential genetic links with (1) differentially spliced genes, (2) differentially expressed transcripts, and (3) spliceosome genes. Specifically, we used a polygenic model to test whether the known SNP-heritability of OUD was enriched for DNA variants in and around the three aforementioned gene sets. We found that the known SNP-heritability of OUD was not significantly linked to DNA variants in any of these gene sets (see Table 2).

## 4. Discussion

Overall, we found support for a few of the study hypotheses. First, our analyses suggest that the abundance of gene isoforms (differentially spliced genes) is different between individuals with OUD compared to the matched controls across brain regions. Building on previous studies, we show that spliceosome genes tend to be upregulated in individuals with OUD relative to the controls, but these changes do not survive stringent significance cut-offs for differential expression. Lastly, we found no support for our hypothesis. Such that, genetic variation in and around differentially spliced or differentially expressed genes was not linked to the heritability of OUD.

Alternative mRNA associations with OUD in the brain may highlight novel neurobiological adaptations to opioid addiction. Differentially spliced OUD genes were linked to both general and brain-related processes in the dlPFC, NAc, and midbrain. The *BIN1* gene (bridging integrator 1, or amphiphysin 2) was differentially spliced in the clathrin and AP2-binding (CLAP) domain of *BIN1* [36] across all samples and brain regions. *BIN1* is highly expressed in nerve terminals and was implicated in clathrin-mediated endocytosis processes enriched in the NAc and dlPFC. Similar genes in the endocytosis pathway were differentially spliced between OUD and the controls and have been functionally characterized to modulate membrane fusion machinery [37], alter intracellular membrane trafficking (*DMN2* or *DYN2*) [38], and impact the structural integrity of vesicles (*CLTA*) [39]. While these genes are not classically studied in opioid use, receptor endocytosis is thought to play an important role in opioid tolerance [40], withdrawal [41], and adaptations to chronic opioid use [42,43].

Many reasons may underlie the differentially spliced genes between OUD and controls. The spliceosome is one critical system underlying the regulation of gene splicing. Our analyses suggest that genes in the spliceosome pathway are upregulated in the brains of individuals with OUD. The most consistent effects in the spliceosome occurred in the U1 and U2 machinery genes. The U1 and U2 components of the spliceosome contain many small nuclear RNAs and mainly function to remove introns from pre-cursor mRNAs [44]. Notably, spliceosomal dysregulation is only one explanation, since observed alternative mRNA splicing changes could also be regulated at the chromatin, translational, or post-translational levels.

Experimental follow-up of correlative human findings are crucial to obtain a deep biological understanding. While individual gene isoforms may lack complete conservation across species, spliceosome genes and the processes of endocytosis are highly conserved across mammals and other model organisms from insects to yeast [45]. These processes influence the fundamental way in which molecular machinery functions in brain cells. More evidence is needed to determine their role in opioid use.

Similar to other studies, we found no significant link between molecular brain readout (differentially expressed or differentially spliced genes) and the heritability of OUD [2]. The genetics of OUD are complex and polygenic. Nonetheless, only one gene—*OPRM1*—has been significantly associated with OUD via GWAS to date, which accounts for a tiny fraction of the total genetic variance. Our model captured 4–6% of the known SNP-heritability in OUD, which was not more than what would be expected by chance. A few scenarios could explain our findings. One reason could be that the current OUD GWAS is too underpowered or that the sample ascertainments between studies are too discrepant to adequately test our hypotheses. Another explanation is that RNA associations with OUD in the brain are not linked to genetics and are either induced directly by drugs or are attributed to other factors increasing noise in our model (genes linked to overdose, different environments, etc.).

The findings of our study should be noted in the context of the following limitations. First and foremost, all results are correlational and our study does not contain any validation or experimental follow-up data. Individuals with OUD have high rates of polysubstance abuse and psychiatric co-morbidities and human post-mortem brain data are limited in their ability to parse the specific components underlying the complex nature of the syndrome. Nonetheless, our study implicates a pervasive and robust role of alternative mRNA splicing in addiction neurocircuitry for humans diagnosed with OUD. These findings warrant future studies investigating the functional significance of alternative splicing on opioid use.

## Figures and Tables

**Figure 1 genes-13-01045-f001:**
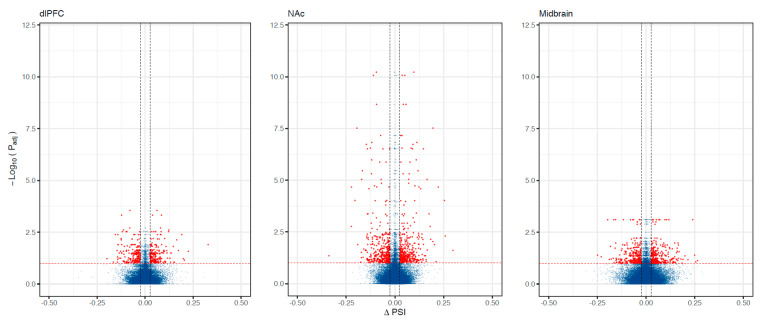
Alternative splicing associations with OUD across brain regions. Volcano plots showing differentially spliced genes associated with OUD. The X-axis shows ΔPSI (splicing effect size) and the y-axis indicates the level of significance (−log_10_ P_adj_). Each dot corresponds to a specific splicing event within a gene. Red dots are significant (P_adj_ < 0.10 and |ΔPSI| > 0.025).

**Figure 2 genes-13-01045-f002:**
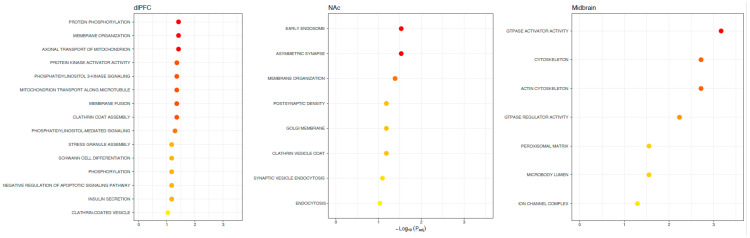
Functions of alternative splicing associations with OUD across brain regions. Possible functions of differentially spliced genes associated with OUD. The X-axis denotes the level of significance and the y-axis shows the different gene ontology terms (KEGG pathways, biological processes, molecular functions, and cellular components). Note no significant (FDR < 0.1) gene ontology terms were linked across all brain regions, highlighting regional specificity.

**Figure 3 genes-13-01045-f003:**
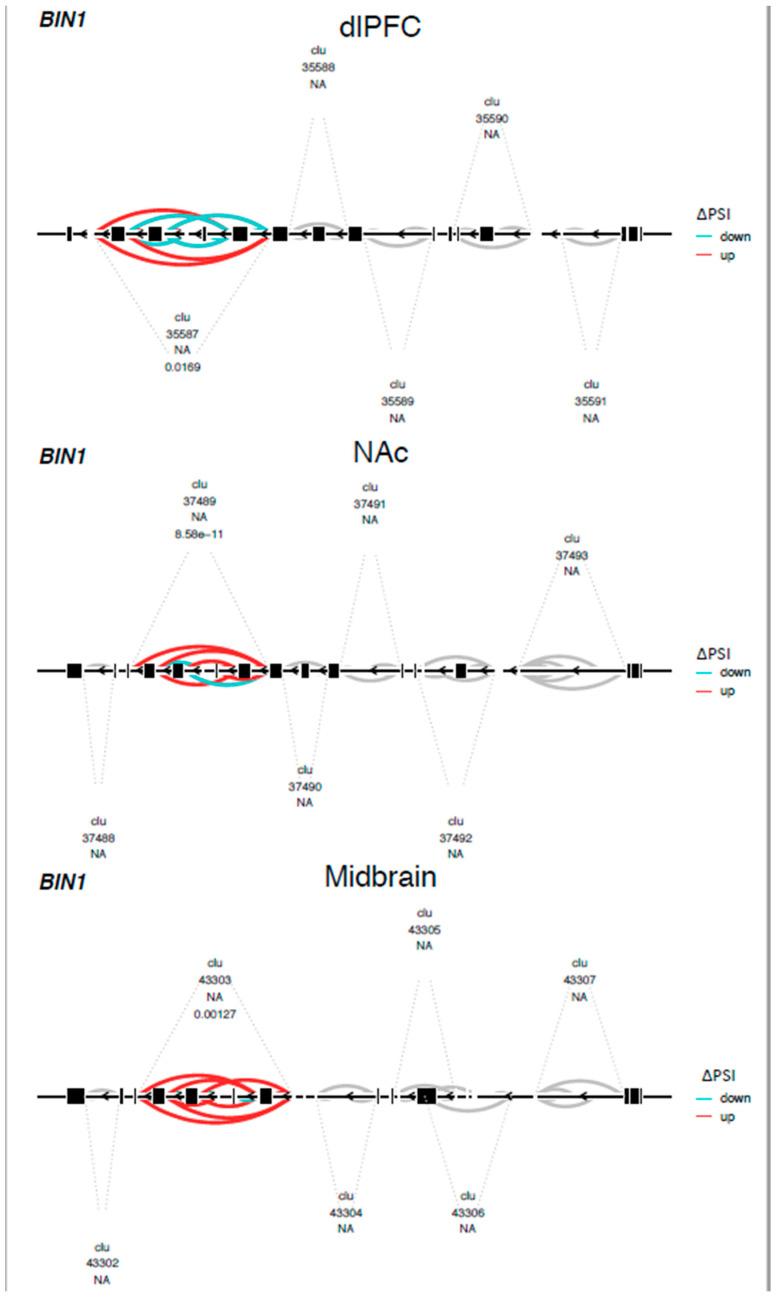
Differential splicing of the BIN1 gene in the CLAP domain across brain regions. Leafviz plot of *BIN1* gene showing a complex splicing event associated with OUD across brain regions. Red indicates increased and blue represents decreased intron usage in individuals with OUD relative to the controls. For a list of specific differential splicing events and their functional characterizations, see Appendix A. Clu = cluster. The number directly below it is an arbitrary cluster number. The number below NA is the FDR-adjusted *p*-value for a cluster.

**Figure 4 genes-13-01045-f004:**
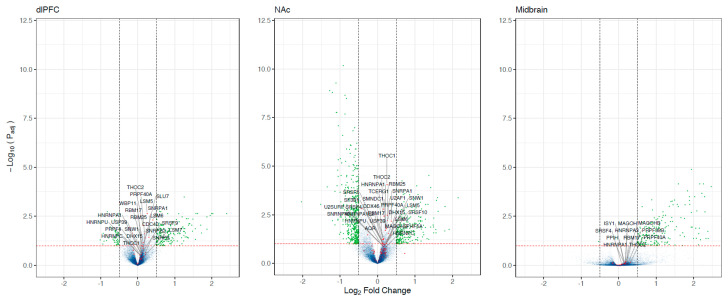
Spliceosome disrupted in the brains of individuals with OUD. A) Volcano plots showing results from differential *expression* analyses of OUD vs. the controls. The X-axis shows the Log_2_ Fold change and the y-axis indicates the level of significance log_10_ (P_adj_). Each dot corresponds to a transcript. Green dots are significant (P_adj_ < 0.10 and |Log_2_ Fold Change| > 0.5), red dots are spliceosome genes. The most significant spliceosome genes are labeled.

**Figure 5 genes-13-01045-f005:**
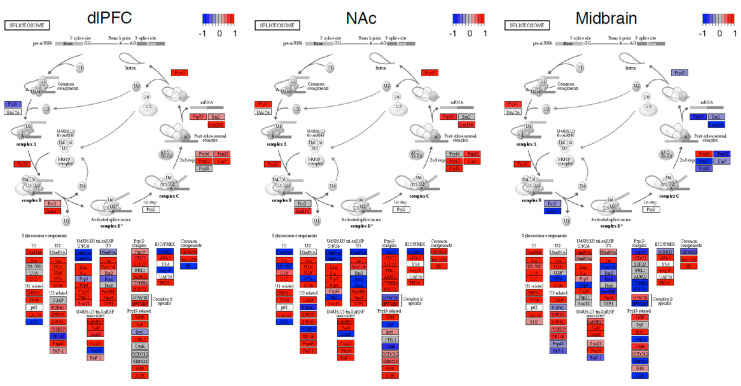
Spliceosome pathway in OUD neurocircuitry. KEGG pathview plots of the spliceosome pathway organized into specific components and stages of the spliceosome. Red denotes an increased expression and blue denotes a decreased expression for a gene in OUD. The −1 to +1 scale for these plots uses a differential expression (Wald) statistic.

**Table 1 genes-13-01045-t001:** Human brain samples of OUD and controls.

Descriptive Information on the RNA-Seq Brain Data				
Variable	DlPFC and NAc [2]		Midbrain [3]	
	OUD	Control	OUD	Control
Sex	50% Female	50% Female	100% Female	100% Female
Age: *M* (s.d.)	46.9 (7.3)	47.3 (9.5)	49.5 (6.4)	52.9 (2.0)
Race	5% African-American	35% African-American	76.66% African-American	60% African-American
RIN: *M* (s.d.)	7.8 (0.7)	8.0 (0.7)	7.3 (0.5)	7.4 (0.5)
Brain pH: *M* (s.d.)	6.4 (0.2)	6.6 (0.3)	6.5 (0.2)	6.6 (0.13)

Note. Race in this table was defined via an autopsy report. dlPFC and NAc samples were derived from the same study and included the same individuals with two different brain regions. RIN stands for RNA Integrity.

**Table 2 genes-13-01045-t002:** Genetic analyses of OUD using genes identified from OUD brain data.

Partitioned Heritability of OUD			
	Differential	Spliceosome	Differentially
	Expressed Genes	Genes	Spliced Genes
Number of Genes	922	127	1141
Surrounding Region	100 kb	100 kb	100 kb
% of Total SNPs	6.06%	0.89%	5.71%
% of OUD *h*^2^_SNP_	4.20%	0.27%	6.48%
Enrichment (se) *h*^2^_SNP_	0.69 (0.47)	0.32 (1.01)	1.13 (0.37)
*p*-value *h*^2^_SNP_	0.506	0.488	0.715

Differentially expressed genes were defined as P_adj_ < 0.10 and |log_2_ fold change| > 0.50, spliceosome genes were a part of the KEGG spliceosome pathway and differentially spliced genes were defined as P_adj_ < 0.10 and |ΔPSI| > 0.025.

## Data Availability

The data used in the current study were obtained via approved Data Use Agreements from the publicly accessible databases referenced in the manuscript.

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
