# Peer review of "Opioid Use Disorder and Alternative mRNA Splicing in Reward Circuitry"

_genes, 2022, doi:10.3390/genes13061045_

Round 1

Reviewer 1 Report

In this paper, Spencer B. Huggett and colleagues, using publicly available RNA-sequencing data, investigated the possible role of alternative mRNA splicing in human brain tissue (dorsal-lateral prefrontal cortex, nucleus accumbens and midbrain of 90 individuals with opiate/opioid use disorder (OUD) or matched controls. The results of analyzes provide evidence of a possible role of alternative mRNA splicing in OUD.

It's an interesting and  well-written manuscript. I think the authors presented an interesting research and the whole study seems to be well done methodically. Thus, I have no criticisms in regard to the manuscript content / text.  However,  I have comments concerning Figures. They are all of an insufficient quality. The resolution of images should be improved, even if it is not easy to develop Figures that have been generated by special software. In case of Fig. 4, splitting the Figure into 3 separate ones may help.

Author Response

We appreciate the kind words and helpful points made by the reviewer. Our revised manuscript increased the resolution of our figures and separated Figure 4 into multiple parts.

Reviewer 2 Report

genes-1714179

Opioid Use Disorder and Alternative mRNA Splicing in Reward Circuitry

Spencer B. Huggett, Ami S. Ikeda, John E. McGeary, Karla R. Kaun, Rohan H.C. Palmer

Opiate/Opioid use disorder (OUD) is a chronic relapsing brain disorder. With publicly available RNA-sequencing data, the authors investigated the possible role of alternative mRNA splicing in human brain tissue of 90 individuals with OUD or matched controls. They found a total of 788 differentially spliced genes across brain regions and identified a functionally characterized splicing change in Clathirin and AP-2-binding (CLAP) domain of the Bridging Integrator 1 (BIN1) gene was significantly linked to OUD across all brain regions. They conclude their findings provide evidence of a possible role of alternative mRNA splicing in OUD.

Analyses of alternative splicing in OUD is an interesting theme. However, it is merely identification of alternatively spliced genes in OUD. The authors did not try to validate splicing changes either RT-PCR or western blotting. In addition, I could not read and tell what are written and what are drawn in Figure 4.

Author Response

Given that we used publicly available data & that our lab is purely computational, it is challenging to conduct experimental follow-up. We agree that our paper just identifies genes linked to OUD from the data. Validation is a key next step. We emphasized this to the reader by stating: “all results are correlational and our study does not contain any validation or experimental follow-up data.”

Additionally, we re-vamped figure 4 (& other figures) to increase the size of text as well as increase the resolution of the images.

Reviewer 3 Report

I have no comments, except some technical mistakes in the whole text.

Author Response

Thank you.

Round 2

Reviewer 2 Report

Although the authors revised the manuscript, I still think validation experiments are required. However, I would like to leave the decision for it to the editor.